# Isolating Sources of Disentanglement in VAEs

**Ricky T. Q. Chen, Xuechen Li, Roger Grosse, David Duvenaud**
University of Toronto, Vector Institute
`rtqichen, lxuechen, rgrosse, duvenaud@cs.toronto.edu`

## Abstract

We decompose the evidence lower bound to show the existence of a term measuring the total correlation between latent variables. We use this to motivate the $\beta$-TCVAE (Total Correlation Variational Autoencoder) algorithm, a refinement and plug-in replacement of the $\beta$-VAE for learning disentangled representations, requiring no additional hyperparameters during training. We further propose a principled classifier-free measure of disentanglement called the mutual information gap (MIG). We perform extensive quantitative and qualitative experiments, in both restricted and non-restricted settings, and show a strong relation between total correlation and disentanglement, when the model is trained using our framework.

## 1 Introduction

Learning disentangled representations without supervision is a difficult open problem. Disentangled variables are generally considered to contain interpretable semantic information and reflect separate factors of variation in the data. While the definition of disentanglement is open to debate, many believe a factorial representation, one with statistically independent variables, is a good starting point [1, 2, 3]. Such representations distill information into a compact form which is oftentimes semantically meaningful and useful for a variety of tasks [2, 4]. For instance, it is found that such representations are more generalizable and robust against adversarial attacks [5].

Many state-of-the-art methods for learning disentangled representations are based on re-weighting parts of an existing objective. For instance, it is claimed that mutual information between latent variables and the observed data can encourage the latents into becoming more interpretable [6]. It is also argued that encouraging independence between latent variables induces disentanglement [7]. However, there is no strong evidence linking factorial representations to disentanglement. In part, this can be attributed to weak qualitative evaluation procedures. While traversals in the latent representation can qualitatively illustrate disentanglement, quantitative measures of disentanglement are in their infancy.

In this paper, we:

- show a decomposition of the variational lower bound that can be used to explain the success of the $\beta$-VAE [7] in learning disentangled representations.

- propose a simple method based on weighted minibatches to stochastically train with arbitrary weights on the terms of our decomposition without any additional hyperparameters.

- introduce the $\beta$-TCVAE, which can be used as a plug-in replacement for the $\beta$-VAE with no extra hyperparameters. Empirical evaluations suggest that the $\beta$-TCVAE discovers more interpretable representations than existing methods, while also being fairly robust to random initialization.

- propose a new information-theoretic disentanglement metric, which is classifier-free and generalizable to arbitrarily-distributed and non-scalar latent variables.

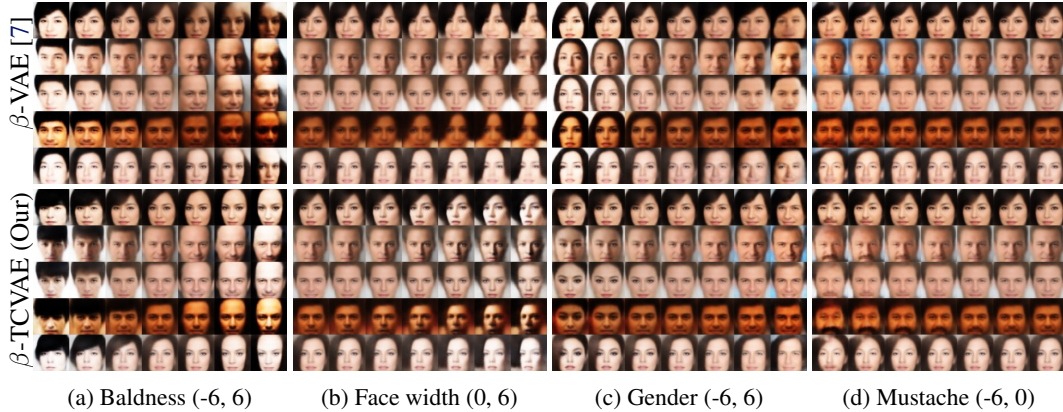

| (a) Baldness (-6, 6) | (b) Face width (0, 6) | (c) Gender (-6, 6) | (d) Mustache (-6, 0) |

Figure 1: Qualitative comparisons on CelebA. Traversal ranges are shown in parentheses. Some attributes are only manifested in one direction of a latent variable, so we show a one-sided traversal. Most semantically similar variables from a $\beta$-VAE are shown for comparison.

While Kim & Mnih [8] have independently proposed augmenting VAEs with an equivalent total correlation penalty to the $\beta$-TCVAE, their proposed training method differs from ours and requires an auxiliary discriminator network.

## 2 Background: Learning and Evaluating Disentangled Representations

We discuss existing work that aims at either learning disentangled representations without supervision or evaluating such representations. The two problems are inherently related, since improvements to learning algorithms require evaluation metrics that are sensitive to subtle details, and stronger evaluation metrics reveal deficiencies in existing methods.

### 2.1 Learning Disentangled Representations

**VAE and $\beta$-VAE**   The variational autoencoder (VAE) [9, 10] is a latent variable model that pairs a top-down generator with a bottom-up inference network. Instead of directly performing maximum likelihood estimation on the intractable marginal log-likelihood, training is done by optimizing the tractable *evidence lower bound* (ELBO). We would like to optimize this lower bound averaged over the empirical distribution (with $\beta = 1$):

$$\mathcal{L}_\beta = \frac{1}{N} \sum_{n=1}^{N} \left( \mathbb{E}_q[\log p(x_n|z)] - \beta \, \text{KL} \left( q(z|x_n) || p(z) \right) \right) \tag{1}$$

The $\beta$-VAE [7] is a variant of the variational autoencoder that attempts to learn a disentangled representation by optimizing a heavily penalized objective with $\beta > 1$. Such simple penalization has been shown to be capable of obtaining models with a high degree of disentanglement in image datasets. However, it is not made explicit why penalizing $\text{KL}(q(z|x)||p(z))$ with a factorial prior can lead to learning latent variables that exhibit disentangled transformations for all data samples.

**InfoGAN**   The InfoGAN [6] is a variant of the generative adversarial network (GAN) [11] that encourages an interpretable latent representation by maximizing the mutual information between the observation and a small subset of latent variables. The approach relies on optimizing a lower bound of the intractable mutual information.

### 2.2 Evaluating Disentangled Representations

When the true underlying generative factors are known and we have reason to believe that this set of factors is disentangled, it is possible to create a supervised evaluation metric. Many have proposed classifier-based metrics for assessing the quality of disentanglement [7, 8, 12, 13, 14, 15].

We focus on discussing the metrics proposed in [7] and [8], as they are relatively simple in design and generalizable.

The Higgins' metric [7] is defined as the accuracy that a low VC-dimension linear classifier can achieve at identifying a fixed ground truth factor. Specifically, for a set of ground truth factors $\{v_k\}_{k=1}^K$, each training data point is an aggregation over $L$ samples: $\frac{1}{L}\sum_{l=1}^L |z_l^{(1)} - z_l^{(2)}|$, where random vectors $z_l^{(1)}, z_l^{(2)}$ are drawn i.i.d. from $q(z|v_k)$[1] for any fixed value of $v_k$, and a classification target $k$. A drawback of this method is the lack of axis-alignment detection. That is, we believe a truly disentangled model should only contain one latent variable that is related to each factor. As a means to include axis-alignment detection, [8] proposes using $\arg\min_j \mathrm{Var}_{q(z_j|v_k)}[z_j]$ and a majority-vote classifier.

Classifier-based disentanglement metrics tend to be *ad-hoc* and sensitive to hyperparameters. The metrics in [7] and [8] can be loosely interpreted as measuring the reduction in entropy of $z$ if $v$ is observed. In section 4, we show that it is possible to directly measure the mutual information between $z$ and $v$ which is a principled information-theoretic quantity that can be used for any latent distributions provided that efficient estimation exists.

## 3    Sources of Disentanglement in the ELBO

It is suggested that two quantities are especially important in learning a disentangled representation [6, 7]: A) Mutual information between the latent variables and the data variable, and B) Independence between the latent variables.

A term that quantifies criterion A was illustrated by an ELBO decomposition [16]. In this section, we introduce a refined decomposition showing that terms describing both criteria appear in the ELBO.

**ELBO TC-Decomposition**    We identify each training example with a unique integer index and define a uniform random variable on $\{1, 2, ..., N\}$ with which we relate to data points. Furthermore, we define $q(z|n) = q(z|x_n)$ and $q(z, n) = q(z|n)p(n) = q(z|n)\frac{1}{N}$. We refer to $q(z) = \sum_{n=1}^N q(z|n)p(n)$ as the *aggregated posterior* following [17], which captures the aggregate structure of the latent variables under the data distribution. With this notation, we decompose the KL term in (1) assuming a factorized $p(z)$.

$$
\mathbb{E}_{p(n)}\Big[\mathrm{KL}\big(q(z|n)||p(z)\big)\Big] = \underbrace{\mathrm{KL}\left(q(z,n)||q(z)p(n)\right)}_{\text{(i) Index-Code MI}} + \underbrace{\mathrm{KL}\big(q(z)||\prod_j q(z_j)\big)}_{\text{(ii) Total Correlation}} + \underbrace{\sum_j \mathrm{KL}\left(q(z_j)||p(z_j)\right)}_{\text{(iii) Dimension-wise KL}}
$$
(2)

where $z_j$ denotes the $j$th dimension of the latent variable.

**Decomposition Analysis**    In a similar decomposition [16], (i) is referred to as the *index-code mutual information* (MI). The index-code MI is the mutual information $I_q(z;n)$ between the data variable and latent variable based on the empirical data distribution $q(z,n)$. It is argued that a higher mutual information can lead to better disentanglement [6], and some have even proposed to completely drop the penalty on this term during optimization [17, 18]. However, recent investigations into generative modeling also claim that a penalized mutual information through the information bottleneck encourages compact and disentangled representations [3, 19].

In information theory, (ii) is referred to as the *total correlation* (TC), one of many generalizations of mutual information to more than two random variables [20]. The naming is unfortunate as it is actually a measure of dependence between the variables. The penalty on TC forces the model to find statistically independent factors in the data distribution. We claim that a heavier penalty on this term induces a more disentangled representation, and that the existence of this term is the reason $\beta$-VAE has been successful.

We refer to (iii) as the *dimension-wise* KL. This term mainly prevents individual latent dimensions from deviating too far from their corresponding priors. It acts as a complexity penalty on the aggregate posterior which reasonably follows from the minimum description length [21] formulation of the ELBO.

We would like to verify the claim that TC is the most important term in this decomposition for learning disentangled representations by penalizing only this term; however, it is difficult to estimate the three terms in the decomposition. In the following section, we propose a simple yet general framework for training with the TC-decomposition using minibatches of data.

A special case of this decomposition was given in [22], assuming that the use of a flexible prior can effectively ignore the dimension-wise KL term. In contrast, our decomposition (2) is more generally applicable to many applications of the ELBO.

## 3.1 Training with Minibatch-Weighted Sampling

We describe a method to stochastically estimate the decomposition terms, allowing scalable training with arbitrary weights on each decomposition term. Note that the decomposed expression (2) requires the evaluation of the density $q(z) = \mathbb{E}_{p(n)}[q(z|n)]$, which depends on the entire dataset[2]. As such, it is undesirable to compute it exactly during training. One main advantage of our stochastic estimation method is the lack of hyperparameters or inner optimization loops, which should provide more stable training.

A naïve Monte Carlo approximation based on a minibatch of samples from $p(n)$ is likely to underestimate $q(z)$. This can be intuitively seen by viewing $q(z)$ as a mixture distribution where the data index $n$ indicates the mixture component. With a randomly sampled component, $q(z|n)$ is close to 0, whereas $q(z|n)$ would be large if $n$ is the component that $z$ came from. So it is much better to sample this component and weight the probability appropriately.

To this end, we propose using a weighted version for estimating the function $\log q(z)$ during training, inspired by importance sampling. When provided with a minibatch of samples $\{n_1, ..., n_M\}$, we can use the estimator

$$\mathbb{E}_{q(z)}[\log q(z)] \approx \frac{1}{M} \sum_{i=1}^{M} \left[ \log \frac{1}{NM} \sum_{j=1}^{M} q(z(n_i)|n_j) \right] \qquad (3)$$

where $z(n_i)$ is a sample from $q(z|n_i)$ (see derivation in Appendix C). This minibatch estimator is biased, since its expectation is a lower bound[3]. However, computing it does not require any additional hyperparameters.

### 3.1.1 Special case: $\beta$-TCVAE

With minibatch-weighted sampling, it is easy to assign different weights $(\alpha, \beta, \gamma)$ to the terms in (2):

$$\mathcal{L}_{\beta-\text{TC}} := \mathbb{E}_{q(z|n)p(n)}[\log p(n|z)] - \alpha I_q(z;n) - \beta \, \text{KL}\big(q(z)||\prod_j q(z_j)\big) - \gamma \sum_j \text{KL}\left(q(z_j)||p(z_j)\right)$$

$$(4)$$

While we performed ablation experiments with different values for $\alpha$ and $\gamma$, we ultimately find that tuning $\beta$ leads to the best results. Our proposed $\beta$-TCVAE uses $\alpha = \gamma = 1$ and only modifies the hyperparameter $\beta$. While Kim & Mnih [8] have proposed an equivalent objective, they estimate TC using an auxiliary discriminator network.

## 4   Measuring Disentanglement with the Mutual Information Gap

It is difficult to compare disentangling algorithms without a proper metric. Most prior work has resorted to qualitative analysis by visualizing the latent representation. Another approach relies on knowing the true generative process $p(n|v)$ and ground truth latent factors $v$. Often these are

semantically meaningful attributes of the data. For instance, photographic portraits generally contain disentangled factors such as pose (azimuth and elevation), lighting condition, and attributes of the face such as skin tone, gender, face width, etc. Though not all ground truth factors may be provided, it is still possible to evaluate disentanglement using the known factors. We propose a metric based on the empirical mutual information between latent variables and ground truth factors.

## 4.1 Mutual Information Gap (MIG)

Our key insight is that the *empirical mutual information* between a latent variable $z_j$ and a ground truth factor $v_k$ can be estimated using the joint distribution defined by $q(z_j, v_k) = \sum_{n=1}^{N} p(v_k) p(n|v_k) q(z_j|n)$. Assuming that the underlying factors $p(v_k)$ and the generating process is known for the empirical data samples $p(n|v_k)$, then

$$I_n(z_j; v_k) = \mathbb{E}_{q(z_j, v_k)} \left[ \log \sum_{n \in \mathcal{X}_{v_k}} q(z_j|n) p(n|v_k) \right] + H(z_j) \tag{5}$$

where $\mathcal{X}_{v_k}$ is the support of $p(n|v_k)$. (See derivation in Appendix B.)

A higher mutual information implies that $z_j$ contains a lot of information about $v_k$, and the mutual information is maximal if there exists a deterministic, invertible relationship between $z_j$ and $v_j$. Furthermore, for discrete $v_k$, $0 \le I(z_j; v_k) \le H(v_k)$, where $H(v_k) = \mathbb{E}_{p(v_k)}[-\log p(v_k)]$ is the entropy of $v_k$. As such, we use the normalized mutual information $I(z_j; v_k)/H(v_k)$.

Note that a single factor can have high mutual information with multiple latent variables. We enforce axis-alignment by measuring the difference between the top two latent variables with highest mutual information. The full metric we call *mutual information gap* (MIG) is then

$$\frac{1}{K} \sum_{k=1}^{K} \frac{1}{H(v_k)} \left( I_n(z_{j^{(k)}}; v_k) - \max_{j \neq j^{(k)}} I_n(z_j; v_k) \right) \tag{6}$$

where $j^{(k)} = \operatorname{argmax}_j I_n(z_j; v_k)$ and $K$ is the number of known factors. MIG is bounded by 0 and 1. We perform an entire pass through the dataset to estimate MIG.

While it is possible to compute just the average maximal MI, $\frac{1}{K} \sum_{k=1}^{K} \frac{I_n(z_{k*}; v_k)}{H(v_k)}$, the gap in our formulation (6) defends against two important cases. The first case is related to rotation of the factors. When a set of latent variables are not axis-aligned, each variable can contain a decent amount of information regarding two or more factors. The gap heavily penalizes unaligned variables, which is an indication of entanglement. The second case is related to compactness of the representation. If one latent variable reliably models a ground truth factor, then it is unnecessary for other latent variables to also be informative about this factor.

As summarized in Table 1, our metric detects axis-alignment and is generally applicable and meaningful for any factorized latent distribution, including vectors of multimodal, categorical, and other structured distributions. This is because the metric is only limited by whether the mutual information can be estimated. Efficient estimation of mutual information is an ongoing research topic [23, 24], but we find that the simple estimator (5) can be computed within reasonable amount of time for the datasets we use. We find that MIG can better capture

| Metric | Axis | Unbiased | General |
|---|---|---|---|
| Higgins *et al.* [7] | No | No | No |
| Kim & Mnih [8] | Yes | No | No |
| MIG (Ours) | Yes | Yes | Yes |

Table 1: In comparison to prior metrics, our proposed MIG detects axis-alignment, is unbiased for all hyperparameter settings, and can be generally applied to any latent distributions provided efficient estimation exists.

subtle differences in models compared to existing metrics. Systematic experiments analyzing MIG and existing metrics are in Appendix G.

## 5   Related Work

We focus on discussing the learning of disentangled representations in an unsupervised manner. Nevertheless, we note that inverting generative processes with known disentangled factors through

weak supervision has been pursued by many. The goal in this case is not perfect inversion but to distill simpler representation [15, 25, 26, 27, 28]. Although not explicitly the main motivation, many unsupervised generative modeling frameworks have explored the disentanglement of their learned representations [9, 17, 29]. Prior to $\beta$-VAE [7], some have shown successful disentanglement in limited settings with few factors of variation [1, 14, 30, 31].

As a means to describe the properties of disentangled representations, factorial representations have been motivated by many [1, 2, 3, 22, 32, 33]. In particular, Appendix B of [22] shows the existence of the total correlation in a similar objective with a flexible prior and assuming optimality $q(z) = p(z)$. Similarly, [34] arrives at the ELBO from an objective that combines informativeness and the total correlation of latent variables. In contrast, we show a more general analysis of the unmodified evidence lower bound.

The existence of the index-code MI in the ELBO has been shown before [16], and as a result, FactorVAE, which uses an equivalent objective to the $\beta$-TCVAE, is independently proposed [8]. The main difference is they estimate the total correlation using the density ratio trick [35] which requires an auxiliary discriminator network and an inner optimization loop. In contrast, we emphasize the success of $\beta$-VAE using our refined decomposition, and propose a training method that allows assigning arbitrary weights to each term of the objective without requiring any additional networks.

In a similar vein, non-linear independent component analysis [36, 37, 38] studies the problem of inverting a generative process assuming independent latent factors. Instead of a perfect inversion, we only aim for maximizing the mutual information between our learned representation and the ground truth factors. Simple priors can further encourage interpretability by means of warping complex factors into simpler manifolds. To the best of our knowledge, we are the first to show a strong quantifiable relation between factorial representations and disentanglement (see Section 6).

# 6   Experiments

We perform a series of quantitative and qualitative experiments, showing that $\beta$-TCVAE can consistently achieve higher MIG scores compared to prior methods $\beta$-VAE [7] and Info-GAN [6], and can match the performance of FactorVAE [8] whilst performing better in scenarios where the density ratio trick is difficult to train. Furthermore, we find that in models trained with our method, total correlation is strongly correlated with disentanglement.[4]

| Dataset | Ground truth factors |
|---------|----------------------|
| dSprites | scale (6), rotation (40), posX (32), posY (32) |
| 3D Faces | azimuth (21), elevation (11), lighting (11) |

Table 2: Summary of datasets with known ground truth factors. Parentheses contain number of quantized values for each factor.

**Independent Factors of Variation**   First, we analyze the performance of our proposed $\beta$-TCVAE and MIG metric in a restricted setting, with ground truth factors that are uniformly and independently sampled. To paint a clearer picture on the robustness of learning algorithms, we aggregate results from multiple experiments to visualize the effect of initialization .

We perform quantitative evaluations with two datasets, a dataset of 2D shapes [39] and a dataset of synthetic 3D faces [40]. Their ground truth factors are summarized in Table 2. The dSprites and 3D faces also contain 3 types of shapes and 50 identities, respectively, which are treated as noise during evaluation.

**ELBO vs.  Disentanglement Trade-off**   Since the $\beta$-VAE and $\beta$-TCVAE objectives are lower bounds on the standard ELBO, we would like to see the effect of training with this modification. To see how the choice of $\beta$ affects these learning algorithms, we train using a range of values. The trade-off between density estimation and the amount of disentanglement measured by MIG is shown in Figure 2.

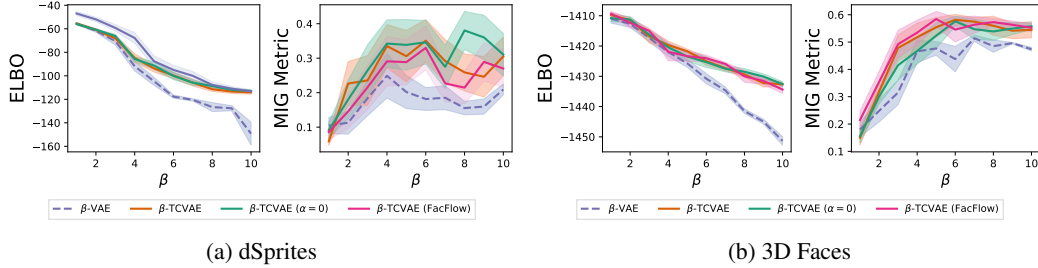

(a) dSprites
(b) 3D Faces

Figure 2: Compared to $\beta$-VAE, $\beta$-TCVAE creates more disentangled representations while preserving a better generative model of the data with increasing $\beta$. Shaded regions show the 90% confidence intervals. Higher is better on both metrics.

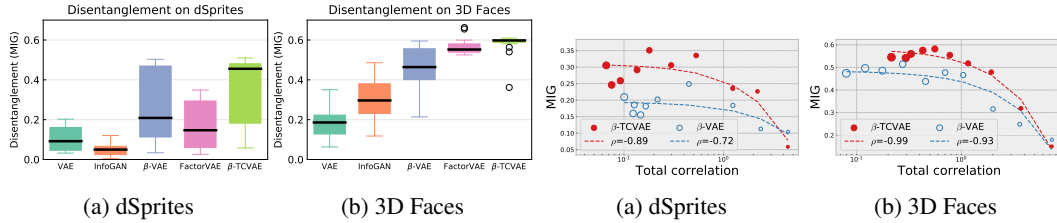

(a) dSprites    (b) 3D Faces    (a) dSprites    (b) 3D Faces

Figure 3: Distribution of disentanglement score (MIG) for different modeling algorithms.

Figure 4: Scatter plots of the average MIG and TC per value of $\beta$. Larger circles indicate a higher $\beta$.

We find that $\beta$-TCVAE provides a better trade-off between density estimation and disentanglement. Notably, with higher values of $\beta$, the mutual information penalty in $\beta$-VAE is too strong and this hinders the usefulness of the latent variables. However, $\beta$-TCVAE with higher values of $\beta$ consistently results in models with higher disentanglement score relative to $\beta$-VAE.

We also perform ablation studies on the removal of the index-code MI term by setting $\alpha = 0$ in (4), and a model using a factorized normalizing flow as the prior distribution which is jointly trained to maximize the modified objective. Neither resulted in significant performance difference, suggesting that tuning the weight of the TC term in (2) is the most useful for learning disentangled representations.

**Quantitative Comparisons**    While a disentangled representation may be achievable by some learning algorithms, the chances of obtaining such a representation typically is not clear. Unsupervised learning of a disentangled representation can have high variance since disentangled labels are not provided during training. To further understand the robustness of each algorithm, we show box plots depicting the quartiles of the MIG score distribution for various methods in Figure 3. We used $\beta = 4$ for $\beta$-VAE and $\beta = 6$ for $\beta$-TCVAE, based on modes in Figure 2. For InfoGAN, we used 5 continuous latent codes and 5 noise variables. Other settings are chosen following those suggested by [6], but we also added instance noise [41] to stabilize training. FactorVAE uses an equivalent objective to the $\beta$-TCVAE but is trained with the density ratio trick [35], which is known to underestimate the TC term [8]. As a result, we tuned $\beta \in [1, 80]$ and used double the number of iterations for FactorVAE. Note that while $\beta$-VAE, FactorVAE and $\beta$-TCVAE use a fully connected architecture for the dSprites dataset, InfoGAN uses a convolutional architecture for increased stability. We also find that FactorVAE performs poorly with fully connected layers, resulting in worse results than $\beta$-VAE on the dSprites dataset.

In general, we find that the median score is highest for $\beta$-TCVAE and it is close to the highest score achieved by all methods. Despite the best half of the $\beta$-TCVAE runs achieving relatively high scores, we see that the other half can still perform poorly. Low-score outliers exist in the 3D faces dataset, although their scores are still higher than the median scores achieved by both VAE and InfoGAN.

**Factorial vs. Disentangled Representations**    While a low total correlation has been previously conjectured to lead to disentanglement, we provide concrete evidence that our $\beta$-TCVAE learning algorithm satisfies this property. Figure 4 shows a scatter plot of total correlation and the MIG

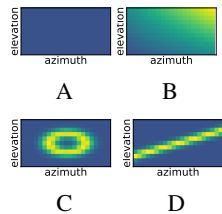

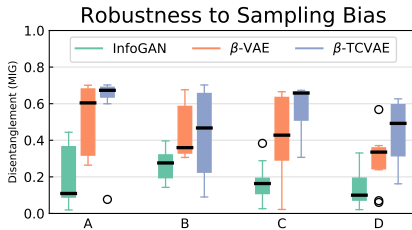

(a) Different joint distributions of factors.

(b) Distribution of disentanglement scores (MIG).

Figure 5: The $\beta$-TCVAE has a higher chance of obtaining a disentangled representation than $\beta$-VAE, even in the presence of sampling bias. (a) All samples have non-zero probability in all joint distributions; the most likely sample is 4 times as likely as the least likely sample.

disentanglement metric for varying values of $\beta$ trained on the dSprites and faces datasets, averaged over 40 random initializations. For models trained with $\beta$-TCVAE, the correlation between average TC and average MIG is strongly negative, while models trained with $\beta$-VAE have a weaker correlation. In general, for the same degree of total correlation, $\beta$-TCVAE creates a better disentangled model. This is also strong evidence for the hypothesis that large values of $\beta$ can be useful as long as the index-code mutual information is not penalized.

## 6.1 Correlated or Dependent Factors

A notion of disentanglement can exist even when the underlying generative process samples factors non-uniformly and dependently sampled. Many real datasets exhibit this behavior, where some configurations of factors are sampled more than others, violating the statistical independence assumption. Disentangling the factors of variation in this case corresponds to finding the generative model where the latent factors can independently act and perturb the generated result, even when there is bias in the sampling procedure. In general, we find that $\beta$-TCVAE has no problem in finding the correct factors of variation in a toy dataset and can find more interpretable factors of variation than those found in prior work, even though the independence assumption is violated.

**Two Factors**   We start off with a toy dataset with only two factors and test $\beta$-TCVAE using sampling distributions with varying degrees of correlation and dependence. We take the dataset of synthetic 3D faces and fix all factors other than pose. The joint distributions over factors that we test with are summarized in Figure 5a, which includes varying degrees of sampling bias. Specifically, configuration A uses uniform and independent factors; B uses factors with non-uniform marginals but are uncorrelated and independent; C uses uncorrelated but dependent factors; and D uses correlated and dependent factors. While it is possible to train a disentangled model in all configurations, the chances of obtaining one is overall lower when there exist sampling bias. Across all configurations, we see that $\beta$-TCVAE is superior to $\beta$-VAE and InfoGAN, and there is a large difference in median scores for most configurations.

### 6.1.1 Qualitative Comparisons

We show qualitatively that $\beta$-TCVAE discovers more disentangled factors than $\beta$-VAE on datasets of chairs [42] and real faces [43].

**3D Chairs**   Figure 6 shows traversals in latent variables that depict an interpretable property in generating 3D chairs. The $\beta$-VAE [7] has shown to be capable of learning the first four properties: azimuth, size, leg style, and backrest. However, the leg style change learned by $\beta$-VAE does not seem to be consistent for all chairs. We find that $\beta$-TCVAE can learn two additional interpretable properties: material of the chair, and leg rotation for swivel chairs. These two properties are more subtle and likely require a higher index-code mutual information, so the lower penalization of index-code MI in $\beta$-TCVAE helps in finding these properties.

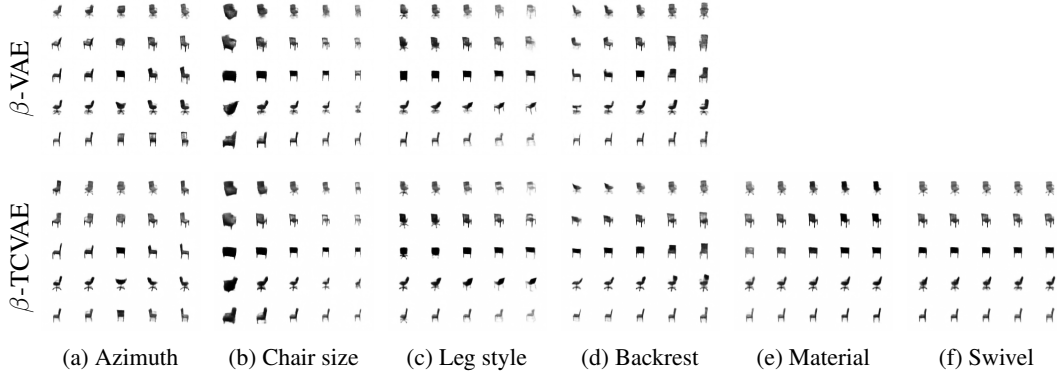

| (a) Azimuth | (b) Chair size | (c) Leg style | (d) Backrest | (e) Material | (f) Swivel |

Figure 6: Learned latent variables using $\beta$-VAE and $\beta$-TCVAE are shown. Traversal range is (-2, 2).

**CelebA** Figure 1 shows 4 out of **15** attributes that are discovered by the $\beta$-TCVAE without supervision (see Appendix A.3). We traverse up to six standard deviations away from the mean to show the effect of generalizing the represented semantics of each variable. The representation learned by $\beta$-VAE is entangled with nuances, which can be shown when generalizing to low probability regions. For instance, it has difficulty rendering complete baldness or narrow face width, whereas the $\beta$-TCVAE shows meaningful extrapolation. The extrapolation of the gender attribute of $\beta$-TCVAE shows that it focuses more on gender-specific facial features, whereas the $\beta$-VAE is entangled with many irrelevances such as face width. The ability to generalize beyond the first few standard deviations of the prior mean implies that the $\beta$-TCVAE model can generate rare samples such as *bald or mustached females*.

## 7 Conclusion

We present a decomposition of the ELBO with the goal of explaining why $\beta$-VAE works. In particular, we find that a TC penalty in the objective encourages the model to find statistically independent factors in the data distribution. We then designate a special case as $\beta$-TCVAE, which can be trained stochastically using minibatch estimator with no additional hyperparameters compared to the $\beta$-VAE. The simplicity of our method allows easy integration into different frameworks [44].To quantitatively evaluate our approach, we propose a classifier-free disentanglement metric called MIG. This metric benefits from advances in efficient computation of mutual information [23] and enforces compactness in addition to disentanglement. Unsupervised learning of disentangled representations is inherently a difficult problem due to the lack of a prior for semantic awareness, but we show some evidence in simple datasets with uniform factors that independence between latent variables can be strongly related to disentanglement.

## Acknowledgements

We thank Alireza Makhzani, Yuxing Zhang, and Bowen Xu for initial discussions. We also thank Chatavut Viriyasuthee for pointing out an error in one of our derivations. Ricky would also like to thank Brendan Shillingford for supplying accommodation at a popular conference.

## Footnotes

[1]Note that $q(z|v_k)$ is sampled by using an intermediate data sample: $z \sim q(z|x), x \sim p(x|v_k)$.

[2]The same argument holds for the term $\prod_j q(z_j)$ and a similar estimator can be constructed.

[3]This follows from Jensen's inequality $\mathbb{E}_{p(n)}[\log q(z|n)] \leq \log \mathbb{E}_{p(n)}[q(z|n)]$.

[4]Code is available at `https://github.com/rtqichen/beta_tcvae`.

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
