[Supplementary Material]

# Appendix for Isolating Sources of Disentanglement in Variational Autoencoders

## A. Random Samples

### A.1 Qualitative Samples

dSprites ($64 \times 64$)　　　　3D Faces ($64 \times 64$)

3D Chairs ($64 \times 64$)　　　　CelebA ($64 \times 64$)

Figure S1: Real samples from the training data set.

## A.3 CelebA Latent Traversals

### $\beta$-TCVAE Model One ($\beta$=15)

Baldness        Dramatic masculinity        Azimuth

Contrast          Mustache (shared with Glasses)          Glasses (shared with Mustache)

Smile (shared with Shadow)        Shadow (shared with Smile)        Gender

Skin color          Brightness          Bangs (side)

Hue            Face width            Eye shadow

## B. Mutual Information Gap

**B.1 Estimation of $I(z_k; v_k)$**

With any inference network $q(z|x)$, we can compute the mutual information $I(z; v)$ by assuming the model $p(v)p(x|v)q(z|x)$. Specifically, we compute this for every pair of latent variable $z_j$ and ground truth factor $v_k$.

We make the following assumptions:

- The inference distribution $q(z_j|x)$ can be sampled from and is known for all $j$.
- The generating process $p(n|v_k)$ can be sampled from and is known.
- Simplifying assumption: $p(v_k)$ and $p(n|v_k)$ are quantized (ie. the empirical distributions).

We use the following notation:

- Let $\mathcal{X}_{v_k}$ be the support of $p(n|v_k)$.

Then the mutual information can be estimated as following:

$$
\begin{aligned}
&I(z_j; v_k) \\
=&\mathbb{E}_{q(z_j,v_k)} \left[\log q(z_j, v_k) - \log q(z_j) - \log p(v_k)\right] \\
=&\mathbb{E}_{q(z_j,v_k)} \left[\log \sum_{n=1}^{N} q(z_j, v_k, n) - \log q(z_j) - \log p(v_k)\right] \\
=&\mathbb{E}_{p(v_k)p(n'|v_k)q(z_j|n')} \left[\log \sum_{n=1}^{N} p(v_k)p(n|v_k)q(z_j|n) - \log q(z_j) - \log p(v_k)\right] \\
=&\mathbb{E}_{p(v_k)p(n'|v_k)q(z_j|n')} \left[\log \sum_{n=1}^{N} \mathbb{1}[n \in \mathcal{X}_{v_k}]p(n|v_k)q(z_j|n)\right] + \mathbb{E}_{q(z_j)} \left[-\log q(z_j)\right] \\
=&\mathbb{E}_{p(v_k)p(n'|v_k)q(z_j|n')} \left[\log \sum_{n \in \mathcal{X}_{v_k}} q(z_j|n)p(n|v_k)\right] + H(z_j)
\end{aligned}
\tag{S1}
$$

where the expectation is to make sampling explicit.

To reduce variance, we perform stratified sampling over $p(v_k)$, and use $10,000$ samples from $q(n, z_k)$ for each value of $v_k$. To estimate $H(z_j)$ we sample from $p(n)q(z_j|n)$ and perform stratified sampling over $p(n)$. The computation time of our estimatation procedure depends on the dataset size but in general can be done in a few minutes for the datasets in our experiments.

**B.2 Normalization**

It is known that when $v_k$ is discrete, then

$$
I(z_j; v_k) = \underbrace{H(v_k)}_{} - \underbrace{H(v_k|z_j)}_{\geq 0} \leq H(v_k)
\tag{S2}
$$

This bound is tight if the model can make $H(v_k|z_j)$ zero, ie. there exist an invertible function between $z_j$ and $v_k$. On the other hand, if mutual information is not maximal, then we know it is because of a high conditional entropy $H(v_k|z_j)$. This suggests our metric is meaningful as it is measuring how much information $z_j$ retains about $v_k$ regardless of the parameterization of their distributions.

## C. ELBO TC-Decomposition

Proof of the decomposition in (2):

$$
\frac{1}{N} \sum_{n=1}^{N} \mathrm{KL}\big(q(z|x_n)||p(z)\big) = \mathbb{E}_{p(n)}\Big[\mathrm{KL}\big(q(z|n)||p(z)\big)\Big]
$$

$$
= \mathbb{E}_{p(n)}\Big[\mathbb{E}_{q(z|n)}\big[\log q(z|n) - \log p(z) + \log q(z) - \log q(z) + \log \prod_j q(z_j) - \log \prod_j q(z_j)]\big]\Big]
$$

$$
= \mathbb{E}_{q(z,n)}\Big[\log \frac{q(z|n)}{q(z)}\Big] + \mathbb{E}_{q(z)}\Big[\log \frac{q(z)}{\prod_j q(z_j)}\Big] + \mathbb{E}_{q(z)}\Big[\sum_j \log \frac{q(z_j)}{p(z_j)}\Big]
$$

$$
= \mathbb{E}_{q(z,n)}\Big[\log \frac{q(z|n)p(n)}{q(z)p(n)}\Big] + \mathbb{E}_{q(z)}\Big[\log \frac{q(z)}{\prod_j q(z_j)}\Big] + \sum_j \mathbb{E}_{q(z)}\Big[\log \frac{q(z_j)}{p(z_j)}\Big]
$$

$$
= \mathbb{E}_{q(z,n)}\Big[\log \frac{q(z|n)p(n)}{q(z)p(n)}\Big] + \mathbb{E}_{q(z)}\Big[\log \frac{q(z)}{\prod_j q(z_j)}\Big] + \sum_j \mathbb{E}_{q(z_j)q(z_{\backslash j}|z_j)}\Big[\log \frac{q(z_j)}{p(z_j)}\Big]
$$

$$
= \mathbb{E}_{q(z,n)}\Big[\log \frac{q(z|n)p(n)}{q(z)p(n)}\Big] + \mathbb{E}_{q(z)}\Big[\log \frac{q(z)}{\prod_j q(z_j)}\Big] + \sum_j \mathbb{E}_{q(z_j)}\Big[\log \frac{q(z_j)}{p(z_j)}\Big]
$$

$$
= \underbrace{\mathrm{KL}\big(q(z,n)||q(z)p(n)\big)}_{\text{(i) Index-Code MI}} + \underbrace{\mathrm{KL}\big(q(z)||\prod_j q(z_j)\big)}_{\text{(ii) Total Correlation}} + \underbrace{\sum_j \mathrm{KL}\big(q(z_j)||p(z_j)\big)}_{\text{(iii) Dimension-wise KL}}
$$

### C.1 Minibatch Weighted Sampling (MWS)

First, let $\mathcal{B}_M = \{n_1, ..., n_M\}$ be a minibatch of $M$ indices where each element is sampled i.i.d. from $p(n)$, so for any sampled batch instance $\mathcal{B}_M$, $p(\mathcal{B}_M) = (1/N)^M$. Let $r(\mathcal{B}_M|n)$ denote the probability of a sampled minibatch where one of the elements is fixed to be $n$ and the rest are sampled i.i.d. from $p(n)$. This gives $r(x_M|n) = (1/N)^{M-1}$.

$$
\mathbb{E}_{q(z)}\left[\log q(z)\right]
$$
$$
= \mathbb{E}_{q(z,n)}\left[\log \mathbb{E}_{n'\sim p(n)}\left[q(z|n')\right]\right]
$$
$$
= \mathbb{E}_{q(z,n)}\left[\log \mathbb{E}_{p(\mathcal{B}_M)}\left[\frac{1}{M}\sum_{m=1}^{M} q(z|n_m)\right]\right]
$$
$$
\geq \mathbb{E}_{q(z,n)}\left[\log \mathbb{E}_{r(\mathcal{B}_M|n)}\left[\frac{p(\mathcal{B}_M)}{r(\mathcal{B}_M|n)}\frac{1}{M}\sum_{m=1}^{M} q(z|n_m)\right]\right] \qquad \text{(S3)}
$$
$$
= \mathbb{E}_{q(z,n)}\left[\log \mathbb{E}_{r(\mathcal{B}_M|n)}\left[\frac{1}{NM}\sum_{m=1}^{M} q(z|n_m)\right]\right]
$$

The inequality is due to $r$ having a support that is a subset of that of $p$. During training, when provided with a minibatch of samples $\{n_1, ..., n_M\}$, we can use the estimator

$$
\mathbb{E}_{q(z)}[\log q(z)] \approx \frac{1}{M}\sum_{i=1}^{M}\left[\log \sum_{j=1}^{M} q(z(n_i)|n_j) - \log(NM)\right] \qquad \text{(S4)}
$$

where $z(n_i)$ is a sample from $q(z|n_i)$.

### C.2 Minibatch Stratified Sampling (MSS)

In this setting, we sample a minibatch of indices $B_M = \{n_1, \ldots, n_m\}$ to estimate $q(z)$ for some $z$ that was originally sampled from $q(z|n^*)$ for a particular index $n^*$. We define $p(B_M)$ to be uniform

over all minibatches of size $M$. To sample from $p(B_M)$, we sample $M$ indices from $\{1, \dots, N\}$ **without replacement**. Then the following expressions hold:

$$
\begin{aligned}
q(z) &= \mathbb{E}_{p(n)}[q(z|n)] \\
&= \mathbb{E}_{p(B_M)}\left[\frac{1}{M}\sum_{m=1}^{M}q(z|n_m)\right] \\
&= \mathbb{P}(n^* \in B_M)\mathbb{E}\left[\frac{1}{M}\sum_{m=1}^{M}q(z|n_m)\Big|n^* \in B_M\right] + \mathbb{P}(n^* \notin B_M)\mathbb{E}\left[\frac{1}{M}\sum_{m=1}^{M}q(z|n_m)\Big|n^* \notin B_M\right] \\
&= \frac{M}{N}\mathbb{E}\left[\frac{1}{M}\sum_{m=1}^{M}q(z|n_m)\Big|n^* \in B_M\right] + \frac{N-M}{N}\mathbb{E}\left[\frac{1}{M}\sum_{m=1}^{M}q(z|n_m)\Big|n^* \notin B_M\right]
\end{aligned}
$$
$$(S5)$$

During training, we sample a minibatch of size $M$ without replacement **that does not contain** $n^*$. We estimate the first term using $n^*$ and $M-1$ other samples, and the second term using the $M$ samples that are not $n^*$. One can also view this as sampling a minibatch of size $M+1$ where $n^*$ is one of the elements, and let $B_{M+1}\backslash\{n^*\} = \hat{B}_M = \{n_1, \dots, n_M\}$ be the elements that are not equal to $n^*$, then we can estimate the first expectation using $\{n^*\} \cup \{n_1, \dots, n_{M-1}\}$ and the second expectation using $\{n_1, \dots, n_M\}$. This estimator can be written as:

$$
q(z) \approx f(z, n^*, \hat{B}_M) = \frac{1}{N}q(z|n^*) + \frac{1}{M}\sum_{m=1}^{M-1}q(z|n_m) + \frac{N-M}{NM}q(z|n_M) \tag{S6}
$$

which is unbiased, and exact if $M = N$.

### C.2.1 Stochastic Estimation

During training, we estimate each term of the decomposed ELBO, $\log p(x|z)$, $\log p(z)$, $\log q(z|x)$, $\log q(z)$, and $\log \prod_{j=1}^{K} q(z_j)$, where the last two terms are estimated using MSS. For convenience, we use the same minibatch that was used to sample $z$ to estimate these two terms.

$$
\mathbb{E}_{q(z,n)}[\log q(z)] \approx \frac{1}{M+1}\sum_{i=1}^{M+1}\log f(z_i, n_i, B_{M+1}\backslash\{n_i\}) \tag{S7}
$$

Note that this estimator is a lower bound on $\mathbb{E}_{q(z)}[\log q(z)]$ due to Jensen's inequality,

$$
\begin{aligned}
&\mathbb{E}_{p(B_{M+1})}\left[\frac{1}{M+1}\sum_{i=1}^{M+1}\log f(z_i, n_i, B_{M+1}\backslash\{n_i\})\right] \\
=&\mathbb{E}_{p(B_{M+1})}\left[\log f(z_i, n_i, B_{M+1}\backslash\{n_i\})\right] \\
\leq &\log \mathbb{E}_{p(B_{M+1})}\left[f(z_i, n_i, B_{M+1}\backslash\{n_i\})\right] \\
=&\mathbb{E}_{q(z)}[\log q(z)]
\end{aligned}
$$
$$(S8)$$

However, the bias goes to zero if $M$ increases and the equality holds if $M = N$. (Note that this is in terms of the empirical distribution $p(n)$ used in our decomposition rather than the unknown data distribution.)

### C.2.2 Experiments

While MSS is an unbiased estimator of $q(z)$, MWS is not. Moreover, neither of them is unbiased for estimating $\log q(z)$ due to Jensen's inequality. Take MSS as an example:

$$
\mathbb{E}_{p(n)}\left[\log MSS(z)\right] \leq \log \mathbb{E}_{p(n)}[MSS(z)] = \log q(z) \tag{S9}
$$

We observe from preliminary experiments that using MSS results in performance similar to MWS.

Figure S7: MSS performs similarly to MWS.

## C. Extra Ablation Experiments

We performed some ablation experiments using slight variants of $\beta$-TCVAE, but found no significant meaningful differences.

### C.1 Removing Index-code MI ($\alpha = 0$)

We show some preliminary experiments using $\alpha = 0$ in (4). By removing the penalty on index-code MI, the autoencoder can then place as much information as necessary into the latent variables. However, we find no significant difference between setting $\alpha$ to 0 or to 1, and the setting is likely empirically dataset-dependent. Further experiments use $\alpha = 1$ so that it is a proper lower bound on $\log p(x)$ and to avoid the extensive hyperparameter tuning of having to choose $\alpha$. Note that works claiming better representations can be obtained with low $\alpha$ [S6, S17] and moderate $\alpha$ [S3] both exist.

Figure S8: ELBO vs. Disentanglement plots showing $\beta$-TCVAE (4) but with $\alpha$ set to 0.

### C.2 Factorial Normalizing Flow

We also performed experiments with a factorial normalizing flow (FNF) as a flexible prior. Using a flexible prior is conceptually similar to ignoring the dimension-wise KL term in (2) (ie. $\gamma = 0$ in (4)), but empirically the slow updates for the normalizing flow should help stabilize the aggregate posterior. Each dimension is a normalizing flow of depth 32, and the parameters are trained to maximize the $\beta$-TCVAE objective. The FNF can fit multi-modal distributions. From our preliminary experiments, we found no significant improvement from using a factorial Gaussian prior and so decided not to include this in the paper.

Figure S9: ELBO vs. Disentanglement plots showing the $\beta$-TCVAE with a factorial normalizing flow (FNF).

## C.3 Effect of Batchsize

Figure S10: We used a high batchsize to account for the bias in minibatch estimation. However, we find that lower batchsizes are still effective when using $\beta$-TCVAE, suggesting that a high batchsize may not be necessary. This is run on the 3D faces dataset with $\beta$=6.

## D. Comparison of Best Models

(a) Best $\beta$-TCVAE (MIG: 0.53)     (b) Best $\beta$-VAE (MIG: 0.50)

Figure S11: **MIG is able to capture subtle differences.** Relation between the learned variables and the ground truth factors are plotted for the best $\beta$-TCVAE and $\beta$-VAE on the dSprites dataset according to the MIG metric are shown. Each row corresponds to a ground truth factor and each column to a latent variable. The plots show the relationship between the latent variable mean versus the ground truth factor, with only active latent variables shown. For position, a color of blue indicates a high value and red indicates a low value. The colored lines indicate object shape with red being oval, green being square, and blue being heart. Interestingly, the latent variables for rotation has 2 peaks for oval and 4 peaks for square, suggesting that the models have produced a more compact code by observing that the object is rendered the same for certain degrees of rotation.

## E. Invariance to Hyperparameters

Figure S12: The distribution of each classifier-based metric is shown to be extremely dependent on the hyperparameter $L$. Each colored line is a different VAE trained with the unmodified ELBO objective.

We believe that a metric should also be invariant to any hyperparameters. For instance, the existence of hyperparameters in the prior metrics means that a different set of hyperparameter values can result in different metric outputs. Additionally, even with a stable classifier that always outputs the same accuracy for a given dataset, the creation of a dataset for classifier-based metrics can still be problematic.

The aggregated inputs used by [S7] and [S8] depend on a batch size $L$ that is difficult to tune and leads to inconsistent metric values. In fact, we empirically find that these metrics are most informative with a small $L$. Figure S12 plots the [S7] metric against $L$ for 20 fully trained VAEs. As $L$ increases, the aggregated inputs become more quantized. Not only does this increase the accuracy of the metric,

but it also *reduces the gap between models*, making it hard to discriminate similarly performing models. The relative ordering of models is also not preserved with different values.

## F. MIG Traversal

To give some insight into what MIG is capturing, we show some $\beta$-TCVAE experiments with scores near quantized values of MIG. In general, we find that MIG gives low scores to entangled representations when even just two variables are not axis-aligned. We find that MIG shows a clearer pattern for scoring position and scale, but less so for rotation. This is likely due to latent variables having a low MI with rotation. In an unsupervised setting, certain ground truth rotation values are impossible to differentiate (e.g. 0 and 180 for ovals and squares), so the latent variable simply learns to map these to the same value. This is evident in the plots where latent variables describing rotation are many-to-one. The existence of factors with redundant values may be one downside to using MIG as a scoring mechanism, but such factors only appear in simple datasets such as dSprites.

Note that this type of plot does not show the whole picture. Specifically, only the mean of the latent variables is shown, while the uncertainty of the latent variables is not. Mutual information computes the reduction in uncertainty after observing one factor, so the uncertainty is important but cannot be easily plotted. Some changes in MIG may be explained by a reduction in the uncertainty even though the plots may look similar.

MIG: 0.0169                MIG: 0.0214

**Score near 0.0**. Representations are extremely entangled.

MIG: 0.0988                    MIG: 0.1017

**Score near 0.1**. Representations are less entangled but fail to satisfy axis-alignment.

MIG: 0.2017                    MIG: 0.2029

**Score near 0.2**. Representations are still entangled, but some form is appearing.

MIG: 0.3155          MIG: 0.3271

**Score near 0.3**. Representations look much more disentangled, with some nuances not being completely disentangled.

MIG: 0.4092          MIG: 0.4179

**Score near 0.4**. Representations appear to be axis-aligned but rotation is still entangled and position is not perfect.

MIG: 0.4929                MIG: 0.5033

**Score near 0.5**. Representations appear to be axis-aligned and disentangled. Higher scores are likely reducing entropy (with latent variables appearing closer to a Dirac delta.) To fully match the ground truth, the latent variables would have to be a mixture of Dirac deltas, but such variables would have a high dimension-wise KL with a factorized Gaussian.

## G. Disagreements Between Metrics

Figure S19: **Entangled representations can have a relatively high Higgins metric while MIG correctly scores it low.** (a) The Higgins metric tends to be overly optimistic compared to the MIG metric. (b, c) Relationships between the ground truth factors and the learned latent variables are shown for the top two controversial models, which are shown as red dots. Each colored line indicates a different shape (see Supplementary Materials). (d) Sample traversals for the two latent variables in model (b) that both depend on rotation, which clearly mirror each other.

Before using the MIG metric, we first show that it is in some ways superior to the [S7] metric. To find differences between these two metrics, we train 200 models of $\beta$-VAE with varying $\beta$ and different initializations.

Figure S19a shows each model as a single point based on the two metrics. In general, both metrics agree on the most disentangled models; however, the MIG metric falls off very quickly comparatively.

In fact, the Higgins metric tends to output a inflated score due to its inability to detect subtle differences and a lack of axis-alignment.

As an example, we can look at controversial models that are disagreed upon by the two metrics (Figure S19b). The most controversial model is shown in Figure S19a as a red dot. While the MIG metric only ranks this model as better than 26% of the models, the Higgins metric ranks it as better than 75% of the models. By inspecting the relationship between the latent units and ground truth factors, we see that only the scale factor seems to be disentangled (Figure S19b). The position factors are not axis aligned, and there are two latent variables for rotation that appear to mirror each other with only a very slight difference. The two rows in Figure S19d show traversals corresponding to the two latent variables for rotation. We see clearly that they simply rotate in the opposite direction. Since the Higgins metric does not enforce that only a single latent variable should influence each factor, it mistakenly assigns a higher disentanglement score to this model. We note that many models near the red dot in the figure exhibit similar behavior.

### G.1 More Controversial Models

Each model can be ordered by either metric (MIG or Higgins) such that each model is assigned a unique integer $1 - 200$. We define the most controversial model as $\max_\alpha R(\alpha; Higgins) - R(\alpha; MIG)$, where a higher rank implies more disentanglement. These are models that the Higgins metric believes to be highly disentangled while MIG believes they are not. Figure S21 shows the top 5 most controversial models.

Figure S20: Models as colored dots are those shown in Figure S21.

(a) $(26\%, 75\%)$    (b) $(11\%, 57\%)$    (c) $(25\%, 70\%)$    (d) $(34\%, 78\%)$    (e) $(28\%, 67\%)$

Figure S21: (a-e) The top 5 most controversial models. The brackets indicate the rank of models by MIG and the Higgins metric. For instance, the most controversial model shown in (a) is ranked as better than 75% of model by the Higgins metric, but MIG believes that it is only better than 25% of models.