[Reviews · NeurIPS 2018]

Reviewer 1



This paper studies the question of why disentangled representations emerge from VAE training, and through the insights derived in this analysis, is able to propose a simple method to obtain even more disentangling in the representations. The fact that beta does not need to be carefully tuned seems like a major advantage. The proposed ideas seem quite insightful/useful, the writing is clear, and the empirical evaluation is thorough and convincing. The authors have been able to shed some light on the murky topic of disentangling, and I believe these results will be of broader interest to the ML community. All of the issues/improvements I initially wanted to bring up, I found had already been addressed after reading through the paper a few more times. Minor Comments: The authors mention there being confusion about the relationship between disentanglement and mutual information between latent codes and the observed data. Have you considered studying how beta affects this mutual information? One major problem with VAEs is the posterior collapse phenomenon, which it seems like the beta-TCVAE would only exacerbate, correct? Perhaps the beta-TCVAE can be combined with methods that increase the mutual information between latent codes and the observed data to avoid posterior collapse, thus ensuring a representation that is both highly disentangled and controllable. Response to Author Feedback: I congratulate the authors on a nice paper and recommend they include the reference to "Controllable Semantic Image Inpainting." in their discussion section. I also recommending including one sentence about the posterior collapse phenomenon somewhere in the text.

Reviewer 2



The paper investigate a decomposition of the ELBO for the training of variational autoencoder that exposes a Total Correlation term, a term that penalizes statistical dependences between latent variables. The authors argue that this term is the important one for achieve disentangled latent variables for instance in the beta-VAE. To support their proposal they develop a minibatch-weighted sampling scheme that allows them to easily modify the weight of the Total Correlation term, resulting in what they call beta-TCVAE. In addition, in order to compare their model to previous models in the literature such as the beta-VAE and InfoGAN, they introduce a new metric to quantify the amount of disentanglement of latent variables, a general quantity that can be efficiently estimated that they call Mutual Information Gap (MIG). Both the the new minibatch estimator to train the objective based on the Total Correlation penalty, and the new disentanglement metric seem novel and significant advancements in the field. Besides explaining the way the beta-VAE works, the new beta-TCVAE model consistently outperforms it in terms of the trade-off between density estimation and disentanglement on several benchmarks, such as dSprites, and 3d Faces.

Reviewer 3



The paper continues the line of work that aims to decompose the variational lower bound of VAEs and try to isolate the component that can be tuned to improve disentanglement in the latent space. Here the authors argue for a particular decomposition that leads to a total correlation term and shows better disentanglement by tuning this term. Strengths: 1. As the authors note, a similar formulation appeared in FactorVAE, but stochastic approximation could be much easier to train than the additional discriminator network used in the later.
 2. The approach is better in terms of the tradeoff between reconstruction and disentanglement.
 3. The paper is well written and does a good job of covering related work.
 Weakness/Questions: 1. The authors should comment on the limitations of the Mutual Information Gap metric. Could there be any stability issues in estimating this?
 2. Is the metric reliable in cases where ground truth factors are correlated?
 3. Are there more compelling reasons to use this approach compared to FactorVAE? Post Rebuttal Comments: I thank the authors for the discussion on the MI metric and providing other clarifications. I hope these are included in the main body of the paper.